# Metabolomic Study to Determine the Mechanism Underlying the Effects of *Sagittaria sagittifolia* Polysaccharide on Isoniazid- and Rifampicin-Induced Hepatotoxicity in Mice

**DOI:** 10.3390/molecules23123087

**Published:** 2018-11-27

**Authors:** Xiu-Hui Ke, Chun-Guo Wang, Wei-Zao Luo, Jing Wang, Bing Li, Jun-Ping Lv, Rui-Juan Dong, Dong-Yu Ge, Yue Han, Ya-Jie Yang, Re-Yila Tu-Erxun, Hong-Shuang Liu, Yi-Chen Wang, Yan Liao

**Affiliations:** 1School of Traditional Chinese Medicine, Beijing University of Chinese Medicine, Beijing 100102, China; 18910167771@163.com (X.-H.K.); wangjingyhc@126.com (J.W.); dearlvwei@163.com (B.L.); drjdongruijuan@126.com (R.-J.D.); dongyuge@163.com (D.-Y.G.); han.yue123@163.com (Y.H.); yangyajie9511@163.com (Y.-J.Y.); reyilaaa@163.com (R.-Y.T.-E.); 18811479932@163.com (H.-S.L.); yswangyichen@163.com (Y.-C.W.); 2Beijing Research Institute of Chinese Medicine, Beijing University of Chinese Medicine, Beijing 100029, China; wangcg1119@126.com; 3Chongqing Academy of Chinese Materia Medica, Chongqing 400065, China; loweizao@163.com; 4Beijing Institute of Biomedicine, Beijing 100091, China; lvjp@chinabib.cn

**Keywords:** ultra-performance liquid chromatography-high resolution mass spectrometry (UPLC-HRMS), isoniazid, rifampicin, *Sagittaria sagittifolia* polysaccharide, liver injury

## Abstract

In this study, a non-targeted metabolic profiling method based on ultra-performance liquid chromatography-high resolution mass spectrometry (UPLC-HRMS) was used to characterize the plasma metabolic profile associated with the protective effects of the *Sagittaria sagittifolia* polysaccharide (SSP) on isoniazid (INH)—and rifampicin (RFP)-induced hepatotoxicity in mice. Fourteen potential biomarkers were identified from the plasma of SSP-treated mice. The protective effects of SSP on hepatotoxicity caused by the combination of INH and RFP (INH/RFP) were further elucidated by investigating the related metabolic pathways. INH/RFP was found to disrupt fatty acid metabolism, the tricarboxylic acid cycle, amino acid metabolism, taurine metabolism, and the ornithine cycle. The results of the metabolomics study showed that SSP provided protective effects against INH/RFP-induced liver injury by partially regulating perturbed metabolic pathways.

## 1. Introduction

Tuberculosis is a pandemic chronic infectious disease caused by *Mycobacterium tuberculosis*. Each year, about 9 million people are diagnosed with tuberculosis worldwide, and about 2 million tuberculosis patients die [1]. Isoniazid (INH) and rifampicin (RFP) are the first-line antituberculosis drugs recommended by the World Health Organization. These two drugs are metabolized in the liver, and both INH and RFP (INH/RFP) cause hepatotoxicity [2]. More importantly, RFP increases the hepatotoxicity of INH in a combination therapy involving INH/RFP [3].

INH/RFP causes irreversible damage, including liver failure, cell necrosis, inflammation, and steatosis [4,5,6]. These effects are considered to be mainly associated with oxidative stress [7], since INH was found to induce apoptosis via oxidative stress and to prevent Nrf2 translocation to the nucleus, thereby preventing its cytoprotective effect [8].

The hepatotoxicity of INH is mediated by its curtailment of mitochondrial function. Oxidative stress can damage the inner membrane of mitochondria, leading to energy metabolism disorders of the tricarboxylic acid cycle. Furthermore, INH and its metabolites (e.g., hydrazine) can cause mitochondrial injury, which can lead to mitochondrial oxidative stress and impairment of energy homeostasis [9].

Plant extracts are known to have the ability to alleviate liver injury by suppressing oxidative stress and hepatocyte apoptosis [10]. Particularly, plant polysaccharides can reduce pathological liver tissue damage, oxidative stress, and inflammatory cells induced by INH/RFP [11,12]. *Sagittaria sagittifolia* L. (Alismataceae) is a perennial aquatic herbal plant also known as Jiandaocao and Yanweicao. It is mainly distributed in the Yangtze River Basin in China and in other countries such as India and Japan. It was recorded in the *Compendium of Materia Medica* as a species native to China and is used as medicine and food [13]. *S. sagittifolia* is highly nutritious, has a pleasing taste, and is inexpensive. The extract of *S. sagittifolia* is known to restrain INH/RFP-induced liver injury [14], and the main hepatoprotective component is *S. sagittifolia* polysaccharide (SSP) [15]. However, the protective mechanism of SSP remains unclear.

Metabolomics involves the measurement of small-molecule metabolite profiles and fluxes in biological matrices after genetic modification or exogenous challenges [16]. An ultra-performance liquid chromatography-high resolution mass spectrometry (UPLC-HRMS)-based metabolomics approach offers a lower detection limit, better selectivity, and higher sensitivity. To extend our previous findings concerning the antioxidative function of SSP, in this study, we used UPLC-HRMS to analyze the metabolites in the plasma of a mouse model to elucidate the mechanism of hepatoprotection rendered by SSP against liver injury caused by INH/RFP.

## 2. Materials and Methods

### 2.1. Instruments and Reagents

HPLC-grade methanol and acetonitrile were purchased from Beijing Invoke Technology Co. Ltd. (Beijing, China). Ultrapure water was prepared using a Milli-Q water purification system (Millipore, Bedford, MA, USA). *S. sagittifolia* was collected from Kunming City, Yunnan Province, China. The specimens were identified as corms of *S. sagittifolia* L. in Alismataceae by Professor Xiuli Wang, School of Chinese Materia Medica, Beijing University of Chinese Medicine, China. *S. sagittifolia* is an accepted name in the plant list [17]. 

SSP was prepared using a previously reported procedure that was simplified as follows [15]. The tuber roots of *S. sagittifolia* were dried at 55 °C and powdered. Then, 20 g of the powder was extracted with 200 mL of boiling water three times. The polysaccharides in the filtrate were precipitated fractionally with alcohol, at which point the proteins were removed from the product by the Sevag method [18]. The polysaccharides were further purified using diethylaminoethyl ion exchange cellulose (DEAE-52). The content of polysaccharide in the tuber roots was measured by the phenol-sulfuric acid method [19] at a wavelength of 490 nm and polysaccharide conversion factor (F) of 1.82. The results showed that the polysaccharide content was 34.31%, which was sufficient for subsequent experiments.

### 2.2. Mouse Model and SSP Administration

All animal studies followed the relevant national legislation and local guidelines and were performed at the College of Traditional Chinese Medicine at Beijing University of Chinese Medicine. A total of 24 BALB/c mice (20 ± 2 g) were commercially obtained from the Military Medical Science Academy of the PLA Animal Center (Beijing, China). Animals were caged under the supply of filtered pathogen-free air and provided food and water ad libitum for 7 days. The mice were randomly divided into three groups: control group (6 mice), model group (INH/RFP, 10 mice), and treatment group (INH/RFP + SSP, 8 mice). The treatment group received SSP at a daily dose of 0.8 g/kg body weight by gavage; the control and model groups received the same amount of normal saline by gavage. After 4 h, the model and treatment groups received INH/RFP at a dose of (0.1 + 0.1) g/kg body weight by gavage. The control group was administered the same amount of normal saline by gavage. All groups received intragastric administration once per day for 30 days.

### 2.3. Plasma Sample Preparation

Samples were obtained from all animals 12 h after administration of the last dose; their body weight was measured, and blood was obtained from the eyeball. A portion of the plasma was used to detect the alanine aminotransferase (ALT) and aspartate aminotransferase (AST) content. The remaining plasma was used for the detection of metabolites.

About 150 μL serum and 450 μL acetonitrile were vigorously shaken for 0.5 min, and the mixture was centrifuged at 15,700× *g* (13,000 rpm) for 10 min at 4 °C to precipitate the protein; the supernatant was re-centrifuged under the same conditions. The supernatant was then dried under nitrogen and dissolved in 160 μL mobile phase A (acetonitrile), followed by centrifugation at 15,700× *g* (13,000 rpm) for 10 min at 4 °C. The supernatant was then injected into a sample bottle.

### 2.4. UPLC-HRMS Analysis

For this analysis, the LTQ-Orbitrap XL mass spectrometer and Dionex UltiMate 3000 UHPLC Plus Focused system (Thermo Fisher Scientific, Waltham, MA, USA) were used. Chromatographic separation was performed on an ACQUITY UPLC TM BEH C18 column (2.1 × 100 mm, 1.7 μm; Waters, Milford, MA, USA) maintained at 30 °C. The mobile phase consisted of 0.1% acetonitrile (A) and ultrapure water (B). The elution gradient started at 5% A and increased to 35% at 0–3 min and then to 95% B at 3–8 min, followed by strong elution for 2 min and equilibration for 5 min. The flow rate was set to 0.34 mL/min, and the injection volume was 5 μL. An electrospray ionization source was used in both the positive and negative modes. The optimized conditions were as follows: capillary voltage, 35 V; drying gas flow, 11 L/min; gas temperature, 350 °C. The data were acquired using Fourier transform at high-resolution full scan mode (resolution, 30,000), and MS/MS spectra were collected using data-dependent acquisition.

### 2.5. Histological Analysis

Liver tissue was placed in 10% neutral formalin solution, embedded in paraffin wax, and cut into sections for hematoxylin and eosin (HE) staining. The histological activity scoring criteria of HE staining were as follows. Both lobular degeneration and portal inflammation were scored. Each item was assigned 0 points (no lesion), 1 point (the lesion area was less than 1/3 of the area of hepatic lobule and portal area), 3 points (the lesion area was less than 2/3 of the area of hepatic lobule and portal area, which was more than 1/3), or 4 points (the lesion range was more than 2/3 of the area of hepatic lobule and portal area).

### 2.6. Data Processing

The data obtained using UPLC-HRMS were imported to Sieve 2.1 software (Waltham, MA, USA) and then preprocessed for ion peak identification, screening, peak alignment, and noise filtering. The data matrices obtained were normalized. The pretreated data were analyzed using SIMCA-P13.0 (Umetrics, Umeå, Sweden), principal component analysis (PCA), and orthogonal signal correction, followed by partial least squares (PLS) mehod. The differential metabolites were identified using Compound Discoverer 2.0 (Thermo Scientific, Waltham, MA, USA) and compared with Human Metabolome Database (HMDB). Functional and pathway enrichment analyses using Kyoto Encyclopedia of Genes and Genomes (KEGG) were conducted to determine differential metabolic pathways [20]. Compound Discoverer 2.0 software (Thermo Scientific, Waltham, MA, USA) accurately matches mzCloud spectra of MS1 and MS2 mass spectral data with the appropriate molecular weight. The parameters were MS1 mass accuracy <5 ppm and a score matching threshold greater than or equal to 60 (the higher the matching, the closer the accuracy to 100). The library search algorithm was High Chem Low + High Res. During the search process, isotope matching and deduction of background noise (S/N ratio < 10,000 as cutoff) were performed.

## 3. Results

### 3.1. Protective Effects of SSP on Ultrastructural Liver Damage Induced by INH/RFP

We performed histological activity score with inflammatory infiltration of hepatic lobule and portal area and both lobular degeneration and portal inflammation were scored. The result (Table 1) indicated that the liver injury of the model group is significantly different from that of the control group and the treatment group.

### 3.2. Effects of SSP on ALT and AST Content

Compared with those in the control group, the ALT and AST levels increased significantly in the model group and decreased significantly in the treatment group (Figure 1).

### 3.3. Normalization and Multivariate Statistical Analysis

Our metabolomics platform was used to analyze the statistically important metabolites. Unsupervised PCA was used first as an unbiased statistical method to investigate the general interrelation of groups, and a clear separation was obtained (Figure 2A,B). Next, PLS discriminant analysis (PLS-DA) was used to maximize the differences in metabolic profiles among the three groups, and the metabolites in the biological samples were detected (Figure 2C,D). A good separation and a clear difference in blood metabolic profiles were noted among the groups, suggesting that INH/RFP and SSP might cause significant changes in the related components. The score plots from PLS-DA showed obvious separation among the three groups in both positive (Figure 2E) and negative ion modes (Figure 2F). The total values of R2Y and Q2, respectively, were higher than 0.6, suggesting that the PLS-DA model was reliable and stable and can sufficiently explain the differences between the two groups of samples. In addition, the model further confirmed that after SSP intervention, the metabolic components of the mice had obviously changed. The model was validated by performing permutation tests with 200 iterations; all permuted R2 (cumulative) and Q2 (cumulative) values to the left were lower than the original point to the right, and the blue regression line of Q2 points had a negative intercept, indicating that the original model was valid. The S-plot showed covariance and correlation between the variables and the model, which decreases the risk of false positives in the selection of potential biomarkers (Figure 3A).

### 3.4. Discovery and Identification of Differential Metabolites

To analyze the metabolic data from these three groups in a biologically meaningful manner, enrichment and metabolite topological analyses of the identified metabolites were carried out. An overview of the pathway analysis is shown in Figure 3B. The pathway analysis contained all the matched pathways arranged by *p*-value on the Y-axis, based on the pathway enrichment analysis, and the pathway impact values arranged on the X-axis, based on the pathway topology analysis. Therefore, many metabolic pathways such as the citrate cycle, arginine, and proline metabolism; phenylalanine, tyrosine, and tryptophan biosynthesis; and valine, leucine, and isoleucine biosynthesis were significantly affected. These findings demonstrate that more than 18 metabolic pathways were significantly dysregulated in the three groups. According to the variable importance in projection (VIP) of the PLS-DA model, the parameters for the evaluation of potential biomarkers were scored at VIP > 1 (the expressed sequence selection of these variables is higher than the average level of clustering effect). Next, t-test and compound screening showed marked differences between the metabolites obtained from the control and model groups. Variables showing differences at *p* < 0.05 were considered potential biomarkers, and 14 compounds that matched those included in the HMDB metabolite retrieval were finally screened. Table 2 showed the mass-charge ratio (M/Z), formula and related pathways of metabolites. These compounds were related to the metabolic pathways of fatty acid metabolism, amino acid metabolism, tricarboxylic acid cycle, and the ornithine cycle (Table 2). A heat map was generated to depict all the metabolites that were significantly altered in the dataset (Figure 3C).

### 3.5. Metabolic Pathway Analysis

The potential metabolites shown in Table 2 are involved in various metabolic pathways. We investigated the links between these metabolites using the KEGG database and HMDB and generated a metabolic map (Figure 4). The map suggested that INH/RFP and SSP had an important influence on the energy metabolism of mice. To further evaluate the effect of pre-administration of SSP on the potential biomarkers, one-way ANOVA with Tukey’s post hoc test was performed between the three groups by SPSS software. The relative peak intensity of the 14 metabolites is shown in Figure 4. Compared with those in the model group, nine metabolites including taurine, L-isoleucine, aspartic acid, ornithine, indole-5,6-quinone, succinic acid, adrenaline, palmitic acid, and L-histidine were significantly recovered in the treatment group. The other five metabolites were also reversed in the model group compared with the control group.

## 4. Discussion

*S. sagittifolia* is recorded in ancient Chinese literature as having various effects, such as a warming and nourishing effects on the viscera [21]. Further, its leaf is used to treat insect and snake bites. When cooked, it can promote digestion and absorption and can be used to treat urethritis and beriberi. Modern research shows that *S. sagittifolia* can relieve liver injury caused by cadmium [22], CCl_4_ [23], and INH/RFP [24]. The mechanism of action might be related to anti-lipid peroxidation and enhancement of free radical scavenging activity [14]. In our study, SSP was also found to protect against liver injury caused by INH/RFP (Table 1).

In this study, UPLC-HRMS was used to investigate the plasma metabolic profile of a mouse model and subsequently elucidate the effect of SSP on liver injury. Fourteen potential biomarkers were identified; these were related to disturbances of the metabolism primarily involving fatty acid metabolism, taurine metabolism, the tricarboxylic acid cycle, ornithine cycle, and amino acid metabolism. Among them, amino acid metabolism is very important because the liver is the only organ that can metabolize all amino acids. Based on the current data, the levels of branched-chain amino acids (BCAAs), histidine, methionine, phenylalanine, and tryptophan (aromatic amino acids) were significantly changed. In the model group, BCAAs were downregulated, whereas the other amino acids were upregulated. This finding is consistent with that of a previous study [25]. In the model group, the pathway through which phenylalanine is broken down into fumaric acid and epinephrine was inhibited. In a previous study, phenylalanine hydroxylase activity was inhibited in liver injury [26]. Therefore, the protective effects of SSP on liver injury might be related to enhancement of the metabolism of phenylalanine.

Furthermore, liver damage often leads to changes in amino acid metabolism. Alterations in amino acid metabolism associated with liver disease are characterized by low levels of circulating BCAAs and elevated levels of circulating aromatic amino acids (phenylalanine, tryptophan, and tyrosine) and methionine [27]. BCAAs can inhibit aromatic amino acids (AAAs) crossing the blood–brain barrier, reduce high ammonia levels in the blood, and maintain the nitrogen balance in the body. BCAAs have important functions such as protection against liver injury. Dietary supplementation of BCAAs ameliorated liver fibrosis in a diethyl nitrosamine-induced rat model of hepatocellular carcinoma with liver cirrhosis [28]. In addition, BCAAs can directly exacerbate hepatic lipotoxicity by reducing lipogenesis and inhibiting autophagy in hepatocytes [29]. Leucine was found to reduce fat synthesis in mouse liver fat cells [30]. In this study, BCAAs were downregulated, while AAAs were upregulated in the model group, indicating that liver injury induced by INH/RFP resulted in a decrease in the clearance of phenylalanine and the release of a large amount of phenylalanine into the blood. Thus, SSP could protect the liver by regulating the metabolism of BCAAs.

Taurine is the most abundant free amino acid in animals and has a wide range of biological functions. For example, it can scavenge oxygen free radicals and resist lipid peroxidation. The liver is not only the main site of taurine synthesis but also an important target organ of taurine. Taurine synthesis decreases when inflammation and liver necrosis occur. It is not only a potential marker of liver injury but also a potential therapeutic agent for liver injury. In the liver, taurine binds to bile acids and participates in bile excretion. Taurine has a protective effect against liver injury related to its role in reducing steatosis and lipid peroxidation [31]. In addition, taurine protects isolated hepatocytes against CCl_4_ and hydrazine cytotoxicity; the mechanisms may include the modulation of calcium levels, osmoregulation, and membrane stabilization [32]. In this study, the content of taurine increased in the model group and decreased in the treatment group, indicating that liver injury caused by INH/RFP resulted in a decrease in taurine synthesis, and the decrease resulted in cholestasis. Therefore, SSP could reduce liver damage by regulating taurine synthesis.

The liver plays an important role in the metabolism of fatty acids since it takes up a large proportion of free fatty acids (FFAs) entering the splanchnic bed through the portal vein [33], with only a small fraction of FFAs being taken up by the non-hepatic splanchnic bed. When the metabolism of liver fatty acids is insufficient, the content of FFAs in the blood and their uptake by the liver increase; this might lead to the accumulation of lipids in liver cells, causing cytotoxicity. Under normal circumstances, mitochondria can decompose excessive amounts of fatty acids in cells via fatty acid beta-oxidation and produce ATP, although the electron respiratory chain leads to excessive generation of reactive oxygen species (ROS) [34]. Overproduction of ROS results in damage of the mitochondrial structure and function and thus the function of the electron respiratory chain, further increasing cellular oxidative stress and causing impaired fatty acid catabolism. As a result, fatty acids are deposited in the cytoplasm [35]. Palmitic acid is an important component of FFAs in the blood. However, the deposition of fatty acids causes cellular toxicity. Palmitic acid has been shown to exhibit a dose-dependent cytotoxic effect associated with ROS production, increasing the level of markers associated with apoptosis and necrosis and decreasing albumin production [36]. Some studies suggested that palmitic acid promotes apoptosis by inducing autophagy in hepatocytes [37]. In this study, the content of palmitic acid was higher in the model group, suggesting that INH/RFP caused lipid deposition in the model group mice, which is somewhat evident in Table 1. Hence, palmitic acid might be a potential biomarker of hepatic damage caused by INH/RFP.

Liver cell damage caused by liver disease can affect the elimination of blood ammonia [38], causing high blood ammonia levels and further liver damage [39]; therefore, a vicious cycle is formed. Under normal circumstances, the liver can metabolize ammonia, a toxin that can decrease the level of ATP in the brain and in turn affect nerve function and nerve cell metabolism, via the ornithine cycle to produce urea, which is then excreted through the kidneys. In this study, the level of ornithine and aspartic acid was higher in the treatment group, indicating that the metabolites of ammonia from the ornithine cycle were higher. Thus, SSP might alleviate liver injury by maintaining the function of the ornithine cycle, and ornithine and aspartic acid might also be potential biomarkers of hepatic damage caused by INH/RFP.

Some studies found that INH/RFP caused liver oxidative stress [40,41], which increased superoxide dismutase activity. Subsequently, peroxidative damage of the mitochondrial endometrium was induced, which activated respiratory chain complexes, leading to a decrease in both ATP synthase and oxidative phosphorylation function. These events culminated in reduced ATP yield and energy metabolism disorders of the tricarboxylic acid cycle [42]. The tricarboxylic acid cycle is a metabolic process that occurs in mitochondria and produces ATP to provide energy for the body. Succinic acid and fumaric acid are important intermediates of the tricarboxylic acid cycle, and their expression can be influenced by liver injury. In this study, succinic acid and fumaric acid were downregulated in the model group and upregulated in the treatment group, suggesting that SSP may protect the tricarboxylic acid cycle and thus prevent liver damage caused by INH/RFP.

## 5. Conclusions

In summary, INH/RFP was found to disrupt metabolism related to energy demand. The PLS-DA-mediated classification for the selection and validation of biomarkers revealed 14 metabolites that could be considered as potential biomarkers. The UPLC-HRMS-based metabolomics method was successfully applied to evaluate the protective effect of SSP. When the potential biomarkers were used as screening indexes, SSP was found to restore fatty acid metabolism, taurine and hypotaurine metabolism, amino acid metabolism, the tricarboxylic acid cycle, and the ornithine cycle. Therefore, SSP may be a promising agent to prevent hepatic injury induced by INH/RFP. However, further studies with a larger sample size are warranted to confirm our findings.

## Figures and Tables

**Figure 1 molecules-23-03087-f001:**
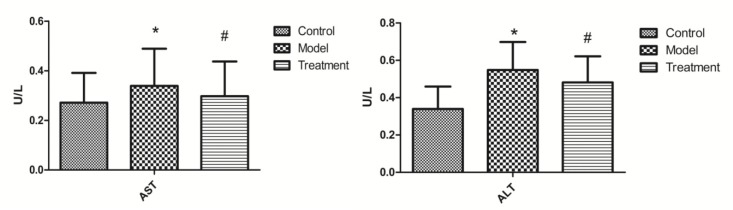
ALT and AST content in mouse plasma. * *p* < 0.01 versus control group; # *p* < 0.05 versus model group.

**Figure 2 molecules-23-03087-f002:**
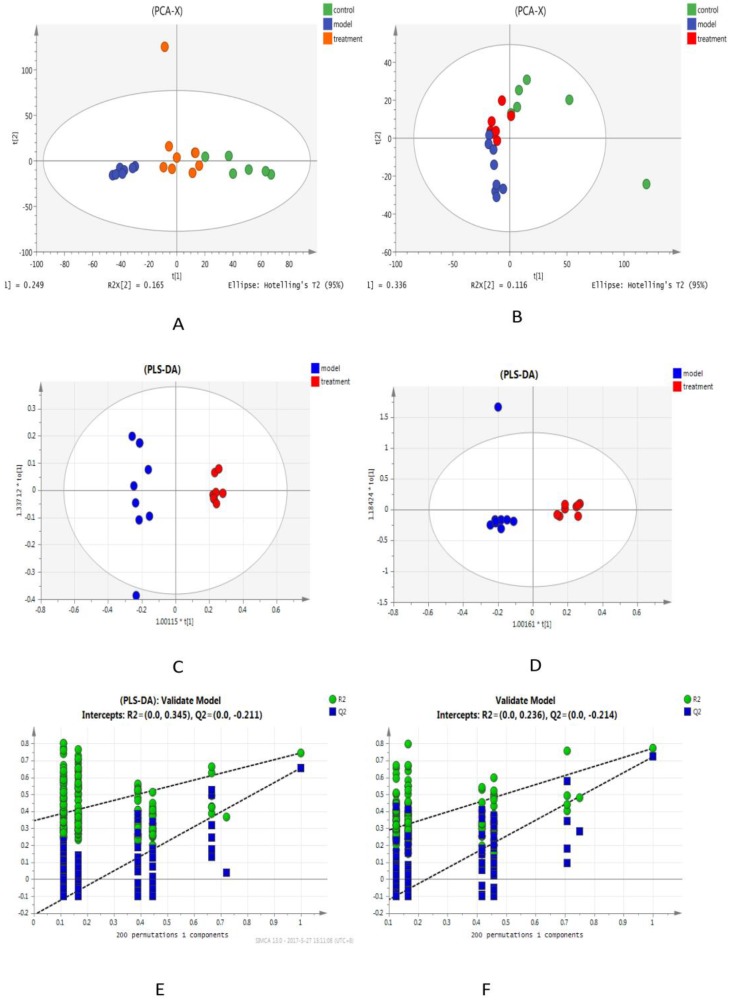
PCA score plot obtained using positive and negative ion mode datasets of the three groups (**A**,**B**) and the model and treatment groups (**C**,**D**). The corresponding permutation test (*n* = 200) of the model and control groups (**E**,**F**). (**A**, **C**, and **E** represent ESI+, and **B**, **D**, and **F** represent ESI-).

**Figure 3 molecules-23-03087-f003:**
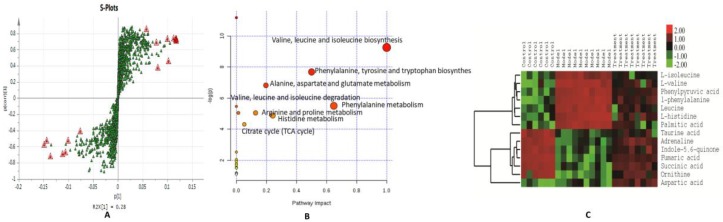
The S-plot obtained using positive and negative ion mode datasets of the control and model groups (**A**). Metabolomic pathway analysis overview indicating selected metabolic pathways that were significantly affected among the control, model, and treatment groups. The size of each node indicates the pathway impact (based on the impact of each identified metabolite in a given pathway). The node color is graded depending on its p-value from pathway enrichment analysis (**B**). We used logarithmic analysis to produce the heat map, showing the 14 significantly altered metabolites based on ANOVA (**C**). Color from green to red represents the intensity of these metabolites from low to high levels in the analysis.

**Figure 4 molecules-23-03087-f004:**
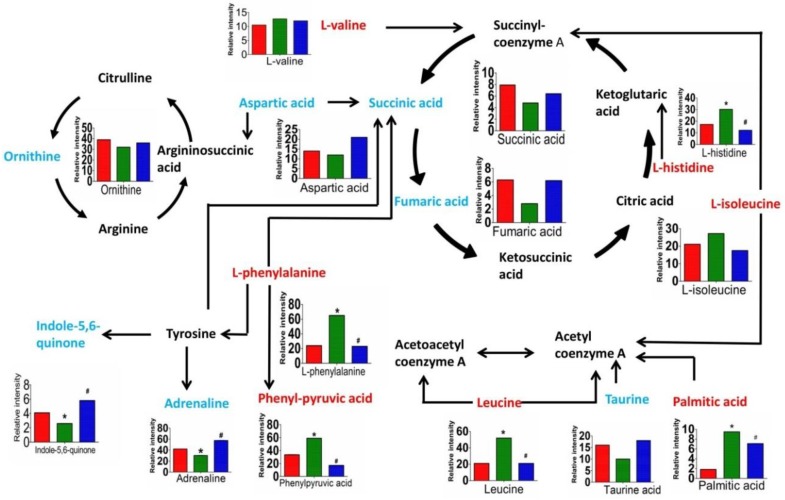
The influence of SSP and INH/RFP was associated with fatty acid metabolism, taurine and hypotaurine metabolism, amino acid metabolism, the tricarboxylic acid cycle, and the ornithine cycle. The different colors of the metabolites represent the following: red, increased in a given group; blue, decreased; and black, not detected. Bar plots show UPLC-HRMS relative signal intensities of the metabolites in the positive and negative modes in the control, model, and treatment groups. In each histogram, red represents the relative expression of the control group, green represents the model group, and blue represents the treatment group. Data are expressed as mean ± SD. * *p* < 0.05 versus control group; # *p* < 0.05 versus model group.

**Table 1 molecules-23-03087-t001:** Histological activity scoring.

		Total Score (Points)		
Groups	n	1	2	3
Control *	6	0	0	0
Model	10	10	0	0
Treatment *	8	0	0	0

* *p* < 0.05 versus model group.

**Table 2 molecules-23-03087-t002:** Summary of signiﬁcantly different metabolites in the mouse plasma of the control, model, and treatment groups.

Time	M/Z	Adduct	Formula	Model Group		VIP	Compound Name	Related Pathway
				Trend	*p*-Value			
12.25	118.0857	M + H	C_5_H_11_NO_2_	↑	1.728 × 10^−3^	1.2453	l-valine	Amino acid metabolism
2.21	126.0217	M + H	C_2_H_7_NO_3_S	↓	1.23 × 10^−2^	1.3751	Taurine acid	
0.89	132.1016	M + H	C_6_H_13_NO_2_	↑	1.737 × 10^−2^	1.4323	l-isoleucine	
2.02	131.0708	M − H	C_6_H_12_O_3_	↑	2.23 × 10^−3^	1.8421	Leucine	
9.08	146.0244	M − H	C_8_H_5_NO_2_	↓	1.32 × 10^−3^	2.0118	Indole-5,6-quinone	
0.58	165.0545	M + H	C_9_H_8_O_3_	↑	1.4712 × 10^−4^	1.4421	Phenylpyruvic acid	
14.14	184.0965	M + H	C_9_H_13_NO_3_	↓	1.08 × 10^−4^	1.2125	Adrenaline	
2.18	164.0711	M − H	C_9_H_11_NO_2_	↑	1.5342 × 10^−2^	1.2421	l-phenylalanine	
1.6	156.0764	M + H	C_6_H_9_N_3_O_2_	↑	1.241 × 10^−4^	1.2232	l-histidine	
2.66	134.0447	M + H	C_4_H_7_NO_4_	↓	1.854 × 10^−3^	1.2524	Aspartic acid	Ornithine cycle
2.66	131.0823	M − H	C_5_H_12_N_2_O_2_	↓	0.92 × 10^−3^	1.7635	Ornithine	
2.05	115.0033	M − H	C_4_H_4_O_4_	↓	0.84 × 10^−3^	1.9302	Fumaric acid	Tricarboxylic acid cycle
1.05	117.0189	M − H	C_4_H_6_O_4_	↓	0.76 × 10^−2^	1.4263	Succinic acid	
3.8	255.232	M − H	C_16_H_32_O_2_	↑	1.22 × 10^−3^	1.4345	Palmitic acid	Fatty acid metabolism

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
