# Peer review of "Metabolomic Study to Determine the Mechanism Underlying the Effects of Sagittaria sagittifolia Polysaccharide on Isoniazid- and Rifampicin-Induced Hepatotoxicity in Mice"

_molecules, 2018, doi:10.3390/molecules23123087_

Round 1

Reviewer 1 Report

This study examined the metabolism underlying the INH/RFP-induced hepatotoxicity and the effect of SSP (Sagittaria sagittifolia polysaccharide), using matabolic profiling method. They showed that SSP is a promissing protective agent against INH/RFP-induced liver injury, restoring fatty acid metabolism, amino acid metabolism. This study is a novel analysis for INH/RFP-induced liver injury, and may provides a new information of approach for treatment. However, there are some minor drawbacks.

1. It is unclear what degree of he INH/RFP –induced liver injury. The authors showed the histology of liver sections. It is unclear. The authors should show the quantification of liver injury, such as necrosis of hepatocytes and infiltaration of neutrophils. The authors also should show the transaminase levels, if serum data is available. 

2. Is this metabolic changes such as amino acids specific for INH/RFP-induced liver injury? For example, in acetoaminophene-induced liver injury, the metabolic changes is same?

Author Response

Response to Reviewer 1 Comments

Point1: It is unclear what degree of the INH/RFP –induced liver injury. The authors showed the histology of liver sections. It is unclear. The authors should show the quantification of liver injury, such as necrosis of hepatocytes and infiltaration of neutrophils. The authors also should show the transaminase levels, if serum data is available. 

Our response: Thanks for valuable suggestions. The quantification of liver injury and transaminase levels were showed in revised manuscript I submitted. In order to show the degree of liver injury, we made statistics on the inflammation classification of pathological pictures. We rated the location of the lesion as the proportion of the acreage of hepatic lobule and portal area. The results showed that although the inflammation score of model group was not high, it was significantly higher than that of control group and treatment group. This indicates that the liver injury of the model group is significantly different from that of the control group and the treatment group. We also showed the level of aminotransferase that we got. As shown in Figure 2, the levels of ALT and AST increased significantly in the model group, indicating that the model group mice had obvious liver injury.

Point2: Is this metabolic change such as amino acids specific for INH/RFP-induced liver injury? For example, in acetoaminophene-induced liver injury, the metabolic changes is same?

Our response: Thank you for your good opinions, which enable us to study more in-depth. Our research demonstrates metabolic changes in INH/RFP-induced liver injury. It is unclear whether these changes are specific, and their specificity needs intensive study to reveal. Although both acetoaminophene and INH/RFP can cause oxidative stress and mitochondrial damage, it is not yet proven that they can cause the same metabolite changes, which needs further study to prove.

Reviewer 2 Report

Comments for the Attention of the Author(s)

    In this manuscript, the authors report “ Metabolomic study to determine the mechanism underlying the effects of Sagittaria sagittifolia polysaccharide on isoniazid and rifampicin-induced hepatotoxicity in mice”. The present investigate the UPLC-HRMS-based metabonomics method was applied to evaluate the protective effect of Sagittaria sagittifolia polysaccharide. The experiments are presented in detail, and the results are carefully discussed.

Comments

1.Figures are hard to follow in current format. The labels and text provided with figures need special attention. Authors must increase font sizes and improve the visibility of data provided.

2. In this Fig.5, the immunohistochemical images of the expression of Nrf2 and HO-1 in liver tissue. This image data was similar to other author paper (1).

Reference:

(1).Wang J, Luo W, Li B, Lv J, Ke X, Ge D, Dong R, Wang C, Han Y, Zhang C, Yu H, Liao Y. Sagittaria sagittifolia polysaccharide protects against isoniazid- and rifampicin-induced hepatic injury via activation of nuclear factor E2-related factor 2 signaling in mice. J Ethnopharmacol. 2018 Dec 5;227:237-245.

3.The discussion needs to be strengthened in terms of possible mode of action and speculating the probable mechanisms for protection.

Author Response

Response to Reviewer2 Comments

Point1: Figures are hard to follow in current format. The labels and text provided with figures need special attention. Authors must increase font sizes and improve the visibility of data provided.

Our response: Thanks for the valuable advice in improving the quality of this paper. The figures, labels and texts have been modified as your request. We revised the definition of figures to improve the quality. The resolution of the figures been modified is higher than 300dpi.

Point2: In this Fig.5, the immunohistochemical images of the expression of Nrf2 and HO-1 in liver tissue. This image data was similar to other author paper (1).

Reference:

(1).Wang J, Luo W, Li B, Lv J, Ke X, Ge D, Dong R, Wang C, Han Y, Zhang C, Yu H, Liao Y. Sagittariasagittifolia polysaccharide protects against isoniazid- and rifampicin-induced hepatic injury via activation of nuclear factor E2-related factor 2 signaling in mice. J Ethnopharmacol. 2018 Dec 5;227:237-245.

Our response: Thanks a lot for the advice. The first authors, Wang Jing and Xiu-Hui Ke, of the two articles are my student both, while the corresponding author is me in the both articles. The same part of experiment was carried out by the same laboratory technician (Dong-Yu Ge and Rui-Juan Dong) in the same platform, only at different time. The similarity of the figures represents the good repeatability of this step of experiment. The previous work was cited in the preface of this article.

Point3: The discussion needs to be strengthened in terms of possible mode of action and speculating the probable mechanisms for protection.

Our response: Thanks for the valuable advice. The discussion about possible mode of action and speculating the probable mechanisms for protection have been revised as your request. We made more supplements to tricarboxylic acid cycle. We supplemented the important role of the tricarboxylic acid cycle in energy metabolism and the regulation of fumaric acid and succinic acid by SSP as intermediates of the tricarboxylic acid cycle. Thus, the regulation effect of SSP on tricarboxylic acid cycle is explained.

Reviewer 3 Report

This metabolomic study was aimed at understanding the mechanism of protection from isoniazid/rifampicin-induced liver injury by the polysaccharide fraction of Sagittaria sagittifolia. The study is well done, however the results are not really convincing : the metabolite profile is not strongly modified by isonoazid/rifampicin treatment, and addition of S. sagittifoliapolysaccharide is sometimes significantly different from the control. We have to bear in mind that the PLS-DA method may discriminate two groups of data easily. The authors should moderate the impact of their results, although they are sufficiently interesting to be published. Finally English should be improved for an easy reading.

Question : Do the authors have informations regarding the nature of the polysaccharide fraction they used ?

Suggestions for minor correctcions in the text :

1.    Lines 35-36 : replace "combination of" by "both" and "causes" by "cause"

2.    Line 40 : replace "as well as" by "and to"

3.    Lines 57 and others : replace "metabonomics" everywhere in the text by "metabolomics" : metabolomics corresponds to the chemical analysis of metabolic profiles, whereas metabonomics mainly describes the comparison of metabolic profiles in human nutrition and pharmaco/toxicology without metabolite identification.

4.    Lines 89 and 91 : replace "was provided" by "received"

5.    Lines 89-90 : what were the modes of administration of INH/RFP and SSP ? Please clarify

6.    Lines 90 and 92 : were the words "injected" appropriate ?

7.    Line 143 : replace "cytoplasmic osteoporosis" by "spongiosis"

8.    Lines 212-214 : replace "the levels of nine metabolites ... were significantly reversed" by "nine metabolites ... significantly recovered"

9.    Line 333 : cancel the word "radioprotective", which means a protection from radiations

Round 2

Reviewer 2 Report

accepted

Author Response

 Thanks.